# Residents and Consultants Have Equal Outcomes When Performing Transrectal Fusion Biopsies: A Randomized Clinical Trial

Beatrice Turchi, Riccardo Lombardo, Antonio Franco, Giorgia Tema, Antonio Nacchia, Antonio Cicione [ID], Antonio Luigi Pastore, Antonio Carbone [ID], Andrea Fuschi, Giorgio Franco, Andrea Tubaro and Cosimo De Nunzio *[ID]

Department of Urology, Sapienza University of Rome, 00100 Rome, Italy; beatrice.turchi@uniroma1.it (B.T.); riccardo.lombardo@uniroma1.it (R.L.); antonio.franco@uniroma1.it (A.F.); g.tema@ospedalesantandrea.it (G.T.); antonio.nacchia@uniroma1.it (A.N.); acicione@libero.it (A.C.); antonioluigi.pastore@uniroma1.it (A.L.P.); antonio.carbone@uniroma1.it (A.C.); andrea.fuschi@uniroma1.it (A.F.); giorgio.franco@uniroma1.it (G.F.); andrea.tubaro@uniroma1.it (A.T.)
* Correspondence: cosimodenunzio@virgilio.it

**Abstract:** The aim of our study was to compare the performance of residents vs. consultants in transrectal fusion prostate biopsies (FUS-PBs), as well as patient-reported comfort. Between January 2021 and October 2022, a consecutive series of patients undergoing FUS-PBs were randomized into two groups: (A) FUS-PBs performed by a consultant; (B) FUS-PBs performed by trained residents (>50 procedures). All patients underwent FUS-PBs with 12 systematic cores and 3/6 target cores. The detection rate and number of positive cores in the target lesion were compared between groups, and the patient's discomfort after the procedure was evaluated using the VAS scale. Overall, 140 patients with a median age of 72 years were enrolled. Overall, 69/140 (49.3%) presented prostate cancer and 53/69 (76.8%) presented a clinically significant cancer (Grade Group $\geq$ 2). Consultants presented a detection rate of 37/70 (52.9%) and residents a detection rate of 32/70 (45.7%) ($p > 0.2$); the mean number of positive cores in the index lesion was similar in both groups (1.5 vs. 1.1; $p > 0.10$). In terms of the patients' experiences, the procedure was well tolerated, with a median VAS score of 2 in both groups, with no statistically significant differences. Residents showed satisfactory outcomes in terms of detection rate, procedural time, and patient comfort when performing prostate biopsies. Residents, after adequate training, can safely perform prostate biopsies.

**Keywords:** prostate cancer; consultants; residents; detection rate; prostate biopsy; target biopsy

## 1. Introduction

Prostate cancer (PCa) is the second most common cancer in men, with more than 1.4 million new diagnoses in 2020 and 375,000 associated deaths worldwide [1]. Prostate cancer should be suspected on the basis of Prostate-Specific Antigen (PSA) levels and/or abnormal Digital Rectal Examination (DRE), but definitive diagnosis depends on histopathological findings after prostate biopsy [2]. Nowadays, since multi-parametric magnetic resonance imaging (mpMRI) of the prostate has become more accessible worldwide, the evaluation of men with elevated PSA levels and clinical suspicion of prostate cancer, and, thus, the decision to perform a prostate biopsy, has drastically changed. Therefore, this imaging technique has begun to play a crucial role in the PCa diagnostic pathway [3–6]. European Guidelines on Prostate Cancer strongly recommend performing an MRI before a prostate biopsy both in the biopsy-naïve subset and in patients with a prior negative biopsy, in order to subsequently perform a prostate biopsy in men with radiological suspicion of PCa [7–9].

Ultrasound (US)- guided or multi-parametric magnetic resonance imaging (mpMRI)-targeted biopsy is currently considered the gold standard in the diagnostic pathway of

PCa [6,10]. Despite US being the most widespread technique, mpMRI demonstrated enhanced sensitivity in the detection of clinically significant prostate cancer, thus making the MRI-targeted biopsy a valuable tool for improved detection of PCa [7,11,12].

To date, although there is still debate on which technique retains the greatest diagnostic ability, three main approaches have been developed for the sampling of suspicious lesions detected with MRI. [13,14] Validated strategies of MRI-guided biopsy include in-bore MRI-targeted biopsy (MRI-TB), which is performed in the MRI suite using real-time MRI guidance without image-fusion technology; MRI-TRUS fusion target biopsy (FUS-TB), which is supported by software that performs an MRI-TRUS image fusion, allowing direct biopsies of MRI-identified lesions; cognitive registration TRUS-targeted biopsy (COG-TB), wherein the MRI images are viewed preceding the biopsy in order to perform a cognitively targeted biopsy of the lesion through TRUS guidance [15]. Among them, considering the widespread use and accessibility of MRI and the latest advancement in image-fusion technology, FUS-TB has the potential to become the diagnostic gold standard for men with suspicion of PCa worldwide [13,16,17]. Nevertheless, as the FUS-TB increases its popularity as a valuable diagnostic tool, performing this approach requires the integration of skills from radiologists and urologists, thus increasing the demand for training in this technique [18]. Young urology residents are nowadays being trained in a setting in which the technology has already taken hold, and they could be the most suitable subjects to develop skills when approaching these new techniques and to implement them in the future. The number of procedures required to reach the plateau of the learning curve and to consider a urology resident fully trained in performing a high-quality prostate biopsy should be considered when evaluating the results of a study comparing procedural outcomes.

Several studies have suggested that experience improves the accuracy of MRI-US-targeted biopsy, and in the assessment of the learning curve, a learning plateau was observed in a range between 50 to 170 procedures [16,18,19].

Indeed, Mager et al. compared the resident's learning curve and the expert's data set, showing that detection rates did not differ significantly between the novice and the expert, whereas biopsy time was demonstrated to be a factor that could be influenced by the level of expertise, resulting significantly longer in the initial novice cohort compared to the expert data set [19]. Similar results were noted by Jia-ao Song et al., who showed no significant difference between residents and a consultant in terms of overall PCa detection rates of systematic biopsy and target biopsy, while it was noted that the consultant had more target biopsy cores and shorter procedural time [20].

An important aspect to take into consideration when assessing prostate biopsy outcomes is the level of comfort experienced by the patient. A transrectal prostate biopsy can be, indeed, a troublesome and painful experience for the patient, since it involves several invasive steps, such as the insertion of the probe, the injection of periprostatic anesthesia, and the sampling of multiple prostate cores. Therefore, it requires adequate expertise directed to minimize patient discomfort and morbidity and to optimize diagnostic accuracy. Although some studies have been performed on this topic, no randomized clinical trials are available on the topic [21].

With this knowledge in mind, the aim of our study is to compare the performance of trained residents and expert consultants when performing transrectal FUS-TB and to assess patient comfort in both scenarios.

## 2. Materials and Methods

### 2.1. Study Population

From January 2021 to October 2022, a consecutive series of patients undergoing transrectal fusion prostate biopsy were randomized into two groups: (A) FUS-PB performed by a consultant; (B) FUS-PB performed by trained residents (>50 procedures). All patients gave written informed consent for biopsy and data collection. Data were prospectively

collected in a single center (Sant'Andrea Hospital of Rome). The study was approved by the local ethics committee: IRU study—Prot. n. 258 SA_2021.

Prostate biopsies were performed when PCa was suspected based on elevated serum PSA values (above 4 ng/mL) or elevated PSA density (above 0.15 ng/mL/cc) and/or abnormal DRE and abnormal mpMRI (PIRADS score $\geq$ 3). Patients were excluded from the study if previous prostate biopsies were performed, if patients were on active surveillance, and if PSA > 30 ng/mL.

Patients were evaluated with a detailed clinical history and a physical examination. Age, prostatic volume evaluated by transrectal ultrasonography (TRUS), and anthropometric parameters, including Body Mass Index (BMI), were recorded. BMI was calculated as weight in kilograms divided by height in meters squared (kg/m$^2$). PSA levels, within a month prior to the biopsy, were registered. Patient discomfort was assessed by rating the level of pain using the Visual Analogue Scale (VAS), consisting of a line with two endpoints representing 0 ("no pain") and 10 ("intense pain"). The questionnaire was submitted to the patient immediately after the procedure.

Patients undergoing FUS-PB were randomized into two groups: (A) FUS-PB performed by an expert consultant; (B) FUS-PB performed by trained residents. Randomization was performed by closed envelopes in a 1:1 ratio. Patients were blinded as well as the clinician performing the data collection and analysis (R.L.). Consultants had more than 10 years of experience and had performed >500 prostate biopsies, while residents were considered "trained" after performing at least 50 procedures prior to the beginning of the study. The training consisted of the first 20 procedures in which the resident performed only the local anesthesia and the random biopsies, while in the following 30, the resident performed the whole procedure. Training was carried out by a senior urologist with more than 10 years of experience. All the biopsies performed by trained residents that we considered in the present study were conducted only after the resident underwent such training; thus, the training bioptic procedures are not included in the study.

All patients underwent a multi-parametric magnetic resonance imaging (mpMRI) prior to the biopsy. All mpMRI was performed using a 1.5 T MRI device and were graded according to the Prostate Imaging Reporting and Data System score, version 2 (PI-RADS v2) [22] by a dedicated radiologist. All the MRI results were then contoured using the bkFusion software (BK5000, BK Medical, DK 2730 Herlev; Denmark), prior to the biopsy, by an expert radiologist, by outlining the prostate shape and each reported lesion, using different signaling colors. Every patient underwent an FUS-TB, according to the mpMRI results [23,24].

FUS-TBs were performed on the patient placed in a left lateral decubitus position. Prior to the procedural date, due to hospital protocols and according to local antimicrobial resistance, patients were administered prophylactic antibiotic therapy with oral sulfamethoxazole and trimethoprim (800 mg + 160 mg; 2 tabs per day) for three days before the biopsy and two days after and amikacin 500 mg iv periprocedurally [25–29]. Povidone–iodine rectal preparation was administered prior to the procedure.

A side-firing transrectal transducer, with its biopsy guide, and a probe-tracking external electromagnetic field generator were used. Preliminary transrectal ultrasound of the prostate was performed, and prostate volume was calculated using the ellipsoid formula. Local peri-prostatic ultrasound-guided anesthesia with 5 mL of 1% mepivacaine (2.5 mL bilaterally) was administered at the apex of the gland using a 22-gauge 15 cm long needle, on both sides of the gland [30]. Previously uploaded MRI prostate contour and TRUS images were fused using the bkFusion software and the contour was manually adjusted on the prostate US image by the operator. An 18-gauge needle was used for prostate cores sampling. All patients underwent combined systematic and targeted biopsy by collecting 12 cores throughout the prostate (parasagittal midline and lateral apical, medial, and basal regions bilaterally) plus 3 to 6 cores for each target lesion [12,31,32].

For each group, the number of positive cores in the target lesions was assessed. Core samples were graded using the International Society of Urological Pathology (ISUP)

Prostate Cancer Grade Group scoring system [33], and PCa was considered clinically significant (csPCa) with a Grade Group $\geq 2$. All specimens were reviewed by a single expert uro-pathologist.

All complications were classified according to the Clavien–Dindo classification system. High-grade complications were defined as Clavien > 3a.

The primary study endpoint was the comparison of prostate cancer detection rates between consultants and trained residents. The number of positive cores sampled in the target lesions, as well as the level of patient comfort experienced during the prostate biopsy, were also evaluated, in order to assess the role of the operator's expertise in performing a prostate biopsy. The time required to perform each procedure was registered and compared in the two groups.

*2.2. Statistical Analysis*

Statistical analysis was performed using the SPSS 23.0 software. Evaluation of the data distribution showed a non-normal distribution of the study data set. Differences between groups of patients in medians for quantitative variables and differences in distributions for categorical variables were tested with Kruskal–Wallis one-way analysis of variance and chi-square test, respectively.

By using multiple logistic regression, the statistically significant variables as assessed in the univariate analysis were entered and investigated as predictors of cancer and csPCa. An alpha value of 5% was considered as the threshold for significance. Data were presented as mean ± standard deviation (SD) and median plus interquartile range (IQR).

Prior to the study start, a power calculation was performed, based on the assumption of an increase in the performance rate of 10% for consultants, when compared to residents, in terms of csPCa detection. In order to identify such a variation, it was estimated that 140 evaluable patients were needed, with 80% power using a two-tailed test at a 5% significance level.

## 3. Results

In total, 140 patients were included in our study and underwent FUS-PBs, of which 70 were performed by a consultant (group 1) and 70 were performed by residents (group 2). The patients' characteristics, as well as clinical data, features of the biopsy (such as number of sampled cores per target lesion, number of positive cores, length of cores in millimeters, procedural time), and health questionnaire results are shown in Table 1. A PIRADS score $\geq 3$ was recorded in 73/140 (52.2%) patients. No significant differences in terms of BMI, prostate volume, PSA levels, and PI-RADS score based on MRI were recorded when comparing pre-biopsy data of patients in group 1 vs. group 2.

Overall, 69 of the 140 patients who underwent an FUS-PB were diagnosed with prostate cancer (49.3%), and 53 of them (76.8%) presented a clinically significant cancer (Grade Group $\geq 2$). The consultants' group presented a detection rate of 37/70 (52.9%), while residents registered a detection rate of 32/70 (45.7%), showing no statistically significant difference between the two groups. Moreover, in terms of patient experience, the procedure was well tolerated by patients, with a median VAS score of 2 in both groups and no statistically significant differences. The time of the procedure was comparable and no significant difference was recorded in terms of core lengths between the two groups.

Overall, we recorded 53/140 (38%) complications. Most of the complications were grade I complications, with hematospermia being the most common complication. Two patients presented a Clavien II complication requiring catheter positioning. No major complications of Clavien $\geq$ IIIa were reported. No significant differences were recorded in terms of complications when comparing both groups (Table 2).

**Table 1.** Overall characteristics of the cohort, MRI data, PB features, and health questionnaire scores.

| Patients (*n*) | 140 |
|---|---|
| Age (years) Median, IQR * | 70.0 (65.0–76.0) |
| PV * (ml) Median, IQR | 52.0 (38.0–72.0) |
| BMI * (kg/m$^2$) Median, IQR | 25.9 (22.5–28.2) |
| PSA * (ng/mL) Median, IQR | 6.8 (5.0–10.2) |
| VAS * Median, IQR | 2 (0–2) |
| PI-RADS * of L1 (1–5) (n) (%) | 140 (100) |
| 1 | 0 (0.0) |
| 2 | 17 (12.1) |
| 3 | 50 (35.7) |
| 4 | 53 (37.9) |
| 5 | 20 (14.3) |
| PI-RADS of L2 (1–5) (n) (%) | 19 (100) |
| 1 | 0 (0.0) |
| 2 | 5 (26.3) |
| 3 | 10 (52.6) |
| 4 | 4 (21.1) |
| 5 | 0 (0.0) |
| Length (mm) Median, IQR | 10 (8–13) |
| Cores for L1 * (n) Median, IQR | 3 (3–5) |
| Cores for L2 * (n) Median, IQR | 3 (3–3) |
| Positive cores L1 (n) Median, IQR | 0 (0–3) |
| Positive cores L2 (n) Median, IQR | 0 (0–2.25) |
| Procedure time (min) Median, IQR | 10 (8–12) |
| PCa (n) (%) | 69 (49.3) |
| ISUP * Grade (1–5) (n) (%) | 69 (100) |
| 1 | 9 (13.0) |
| 2 | 7 (10.1) |
| 3 | 10 (14.5) |
| 4 | 19 (27.5) |
| 5 | 24 (34.8) |

* IQR: interquartile range; PV: prostate Volume; BMI: Body Mass Index; PSA: Prostate-Specific Antigen; VAS: Visual Analogue Scale; PI-RADS: Prostate Imaging Reporting and Data System; L1: index lesion; L2: lesion 2; ISUP: International Society of Urological Pathology.

**Table 2.** Consultants' versus residents' outcomes.

| Variables | Group 1 | Group 2 | P |
|---|---|---|---|
| AGE (years)<br>Median, IQR | 72 (66–78) | 68 (63–73) | 0.001 |
| PV (mL)<br>Median, IQR | 51 (37–68) | 56 (40–84) | 0.106 |
| BMI (kg/m$^2$)<br>Median, IQR | 26.2 (22.8–28.1) | 25.1 (21.8–28.9) | 0.578 |
| PSA (ng/mL)<br>Median, IQR | 6.7 (5.4–9.4) | 7.0 (4.4–10.4) | 0.571 |
| VAS<br>Median, IQR | 2 (0–2) | 2 (0–2.5) | 0.774 |
| PI-RADS of L1 (1–5)<br>(n) (%) | 70 | 70 | |
| 1 | 0/70 (0.0) | 0/70 (0.0) | |
| 2 | 8/70 (11.4) | 9/70 (12.9) | |
| 3 | 17/70 (24.3) | 33/70 (47.1) | 0.013 |
| 4 | 31/70 (44.3) | 22/70 (31.4) | |
| 5 | 14/70 (20.0) | 6/70 (8.6) | |
| PI-RADS of L2 (1–5)<br>(n) (%) | 10 | 9 | |
| 1 | 0 (0.0) | 0 (0.0) | |
| 2 | 4 (40.0) | 1 (11.1) | |
| 3 | 3 (30.0) | 7 (77.8) | 0.758 |
| 4 | 3 (30.0) | 1 (11.1) | |
| 5 | 0 (0.0) | 0 (0.0) | |
| Size (mm) | 8 (6/12) | 9 (7/13) | 0.354 |
| Location (anterior/peripheral) | 10/60 | 12/58 | 0.642 |
| Length (mm)<br>Median, IQR | 9 (8–13) | 10 (9–12) | 0.572 |
| Max cancer length | 5 (3/6) | 4 (3/7) | 0.898 |
| Cores for L1 (n)<br>Median, IQR | 3 (3–3) | 3 (3–5) | 0.923 |
| Cores for L2 (n)<br>Median, IQR | 3 (2–3) | 3 (3–3) | 0.038 |
| Positive cores L1 (n)<br>Median, IQR | 0.5 (0.0- 3.0) | 0 (0–2) | 0.104 |
| Positive cores L2 (n)<br>Median, IQR | 1 (0–3) | 0 (0–0) | 0.135 |
| Procedure time (min)<br>Median, IQR | 10 (8–12) | 9 (8–12) | 0.143 |
| PCa (n) (%) | 37 (52.9) | 32 (45.7) | 0.499 |
| ISUP Grade (1–5)<br>(n) (%) | 37 (100) | 32 (100) | |
| 1 | 3 (8.1) | 6 (18.8) | |
| 2 | 4 (10.8) | 3 (9.4) | |
| 3 | 5 (13.5) | 5 (15.6) | 0.196 |
| 4 | 10 (27) | 9 (28.1) | |
| 5 | 15 (40.5) | 9 (28.1) | |
| Complications<br>(n) (%) | | | |
| Haematospermia | 18/70 (26%) | 20/70 (28%) | 0.345 |
| Haematuria | 5/70 (7%) | 7/70 (10%) | 0.432 |
| Rectal Bleeding | 1/70 | 0/70 | 0.833 |
| Fever > 38.5° | 0/70 | 0/70 | 1.000 |
| AUR * | 1/70 | 1/70 | 1.000 |

\* Acute Urinary Retention.

## 4. Discussion

In the present study, we randomized patients to trained residents and expert urologists to evaluate possible differences in terms of detection rate and patient compliance. According to our results, the performance of trained residents is similar to that of consultants, with no statistically significant differences in terms of PCa and high-grade PCa detection. Although the detection rate was slightly higher in the consultant group, the results were not statistically significant (53% vs. 47%; *p* = 0.499). Similarly, the procedure was well tolerated in both groups, with median VAS scores of 2. The present study is in line with

the peer-reviewed literature and confirms that trained residents can safely perform fusion biopsies with excellent results.

When assessing the effectiveness of learning processes for surgical methods, it is essential to consider a variety of elements including technology features, the nature of the centers, and the characteristics of both the population and the operators. This is particularly true for prostate biopsy procedures, wherein the primary measure of success, cancer detection, lacks a definitive gold standard for testing. This outcome is significantly impacted by the random prevalence and distribution of the disease in the sample group. Our study population highlighted these complexities, especially given the variances between the two participating cohorts. These differences, however, may enhance the applicability of our findings. The existing research presents a mixed and sometimes conflicting view on the learning curve associated with cancer detection in biopsy procedures similar to ours.

Several studies have evaluated the learning curve for FUS-PB. Kasabwala et al., showed that the accuracy of MRI-US-targeted biopsies performed by a single operator in a series of 173 consecutive PBs increased up to the 98th targeted biopsy, in terms of a higher precision in index lesion targeting, an increase in specimen quality with a lower percentage of fibromuscular tissue represented in the samples, and an increase in peripheral zone sampling. Nonetheless, a change in csPCa detection over time was not observed [16]. Additionally, Xu et al., identified a learning phase for targeted FUS-PB as the first 52 cases, after which a consistent cancer detection rate was observed [18]. Furthermore, Mager et al., compared a novice's initial group of 42 FUS-PBs with the 42 subsequent biopsies, by assessing the detection rate and biopsy time as the main parameters of the learning progress and demonstrating that a learning plateau was reached after 63 FUS-PBs [19].

Our study focused on assessing differences in the outcomes of FUS-PBs when performed by consultants or residents. The present study represents the first RCT evaluating FUS-PBs performed by residents versus consultants, confirming the equal outcomes in terms of detection rate and patient satisfaction in both groups. Residents performing FUS-PBs were trained by an expert and had carried out at least 50 procedures prior to the time considered for this study. According to the data in the literature, this could be considered a reasonable number of procedures to establish that the resident had reached the learning curve plateau prior to the beginning of the study, at least when acknowledging the overall PCa detection rate. We considered the PCa detection rate and the number of positive cores in the target lesion to be the main parameters of biopsy quality, and no statistically significant difference was found between the two groups. Data in the literature show, indeed, similar findings.

Few studies have compared the results of prostate biopsies when performed by residents with those performed by consultants. In the comparison of three different trainees and a single consultant, Jia-ao Song et al., assessed no significant difference between the residents and consultant in terms of overall PCa detection rates, while it was noted that the consultant sampled a higher number of target cores and accomplished shorter procedural time [20]. Mager et al., compared the novice's learning curve and the expert's data set, revealing a significant procedural time difference between the groups. The initial novice cohort indicated slower times compared to the expert data set, whereas the PCa detection rate did not differ significantly between the novice and the expert groups [19].

Checcucci et al. reported that after more than a thousand biopsies, the overall PCa and csPCa detection rate was not affected by operator experience. It is interesting to notice that when focusing on lesions <8 mm in diameter, a statistically significant improvement for both PCa and csPCa detection rate was indeed registered parallel to the increase in the number of procedures, and thus, the level of expertise [34]. The present study was not statistically powerful enough to answer this question; however, a study is ongoing to analyze the performance of residents in the detection of smaller lesions.

The study's focus was also directed toward the assessment of the comfort of patients right after undergoing an FUS-PB when performed by consultants or by residents, since prostate biopsy is an invasive and bothersome procedure that holds the potential for

inducing significant pain in the patient. Proper technique in prostate biopsy requires the mastering of several steps, such as probe insertion, periprostatic block needle injection, and sampling of multiple appropriately placed core biopsies. Thus, it is an essential aspect of the adequate training of residents in FUS biopsy that they are directed to minimize patient discomfort and morbidity as well as to optimize diagnostic accuracy.

When comparing patient-reported comfort in transrectal ultrasound-guided biopsies performed by senior urologist-supervised residents and by attending staff urologists alone, CT Nguyen et al. showed that under senior supervision and after a proper amount of training, urology residents can perform the procedure without causing higher levels of discomfort to the patients when compared to those performed by senior urologists [35].

When considering the results of the VAS scale questionnaire, submitted to each patient immediately after the procedure, no statistically significant difference was registered in our study. This shows that the procedure was equally well tolerated by the patients in both groups. The present study is the first study assessing reported outcomes in patients undergoing a prostate biopsy fully performed by a resident, confirming that FUS-PBs may be safely performed by well-trained residents.

Although training is very important for fusion biopsies, it is important to highlight that there are several different software platforms and brands available for fusion biopsies. Software may be categorized into fixed-arm with elastography, fixed-arm, or freehand, and based on real-time images or requiring simulation. When using real-time fixed-arm software, the surgeons have very little to do besides inserting the needle, and therefore, training is quite simple and does not require particular surgical skills. When using real-time transperineal freehand software, the learning curve is probably steeper, considering that the required surgical skills are much more complex to acquire, and the fusion may be altered by the direction of the transrectal probe. In the near future, the implementation of artificial intelligence software to aid the fusion process and to adapt the fusion during the procedure will improve the precision of the retrieval of cores. Likewise, the introduction of robots who may perform biopsies automatically may improve the accuracy of core tracking even more [23]. Although all these are important technological improvements, the accuracy of fusion targeting is severely restricted by the different locations and shapes obtained in the MRI images acquired by the software prior to the procedure and in the TRUS image obtained intra-operatively by the probe. In our study, we used transrectal guided real-time software, which is quite easy to learn for residents, and therefore, our study results may not apply to other software available or for TP techniques.

Notwithstanding all these important areas of debate regarding the accuracy of fusion biopsies, the real issue nowadays lies in the quality of MRI studies and reporting. The overall accuracy of the fusion biopsy clearly depends on the MRI and, therefore, efforts should be made to ensure the centralization of MRI in specialized centers. As stated before, the introduction of automatic reading by AI-aided software may fill the gap. Recently, Sun et al. [36] have demonstrated how on the lesion level, AI-aided MRI reporting enhanced sensitivity from 40.1% to 59.0% (18.9% increase; 95% confidence interval (CI) (11.5, 26.1); $p < 0.001$). On the patient level, AI-aided MRI reporting enhanced the specificity from 57.7 to 71.7% (14.0% increase, 95% CI (6.4, 21.4); $p < 0.001$), while the sensitivity was equal (88.3% vs. 93.9%, $p = 0.06$). AI-aided MRI reporting reduced the median reading time for each case by 56.3%, from 423 to 185 s (238 s decrease, 95% CI (219, 260); $p < 0.001$), while the median diagnostic confidence score was increased by 10.3%, from 3.9 to 4.3 (0.4-score increase, 95% CI (0.3, 0.5); $p < 0.001$) [37]. Finally, the workload should be reduced by accurately selecting patients needing an MRI as well as selecting patients needing fusion prostate biopsies. In our study, to reduce possible biases due to MRI reading, all MRI images were read and contoured by our dedicated uro-radiologist.

Another important area of debate is the indication of transperineal biopsies over transrectal biopsies. According to the EAU guidelines, transperineal should be preferred over transrectal biopsies to improve the cancer detection rate and to reduce the risk of infections and sepsis (sepsis rates are 0.1% and 0.9% for transperineal and transrectal

biopsies, respectively) [7], thus lowering the risk of subsequent hospital re-admission. According to Pepe et al., on a series of 8.500 men undergoing a transperineal prostate biopsy, 1.5% of them required hospital re-admission due to complications but none developed sepsis [7,37]. Notwithstanding the recommendation, the transrectal route is still widely used all over the world. A recent meta-analysis compared the diagnostic precision of transperineal (TP) versus transrectal (TR) MRI-TRUS fusion prostate biopsy (PBx) and found comparable sensitivity and specificity in identifying clinically significant prostate cancer (CSPCa) using both methods. Nevertheless, significant variability was noted among the studies [38]. Rai et al., also conducted a similar analysis and found that CSPCa detection was notably higher with the transperineal approach compared to the transrectal method. However, they described the evidence as having "very low" certainty, highlighting the lack of comprehensive data, and underscoring the need for further research in this area [39]. Furthermore, Kaneko et al., recently performed a match-paired analysis to directly compare fusion TP vs. fusion TR biopsies. According to their results, transperineal MRI-TRUS fusion biopsies provide similar detection rates for CSPCa, with higher positive core lengths and percent of core involvement than transrectal biopsies [40]. A recent systematic review from Uleri et al. found no statistically significant difference in the detection of csPCa between TR and TP approaches but, when stratified by lesion location, the TP approach was associated with higher csPCa detection of anterior and apical lesions [41]. Based on the available evidence, sources of bias when looking at different cohorts of patients clearly limit the conclusions of the available meta-analysis. Although fixed-arm, transperineal, real-time fusion machines with AI and elastography programs represent the future gold standard, time and costs clearly limit the widespread use of these machines. Finally, some authors clearly support the widespread use of cognitive fusion. Pirola et al. recently performed a systematic review and meta-analysis to compare cognitive vs. software fusion biopsy. According to the eight studies evaluated, the detection rates of csPCa were similar between the two groups (OR 1.01, 95% CI 0.74–1.37, $p = 0.95$) and study heterogeneity was low ($I^2$ 55%) [42]. The literature on the subject is very vast, and hundreds of papers are being published on the topic. However, the abovementioned biases in the interpretation of MRI and in the performance of biopsies limit the conclusions. In our center, we perform TR prostate biopsies with antibiotic prophylaxis and povidone–iodine rectal cleansing with optimal detection rates, and septic events are anecdotally reported.

Some limitations must be acknowledged in our study. The ideal design to compare the detection rate between both groups would have been to biopsy the same patients in both groups. However, such a design presents important technical and ethical issues. The present study confirms that trained residents reach comparable results when compared to consultants; however, it is important to underline the importance of surgical expertise. A study is ongoing in our center to explore the importance of surgical expertise in patients with small PIRADS 4 lesions (<10 mm). Small lesions in the base of the prostate located peripherally represent a challenge for most clinicians performing fusion biopsies. The assessment of the learning curve could be an arduous task as numerous parameters should be evaluated in order to investigate the trend of the learning progress. At our center, residents were trained prior to this study following our institution's training protocol; thus, the possibility of different urological training within other urological institutions was not accounted for. Secondly, in the assessment of the performance of FUS-PBs and, thus, of the learning curve as well, multiple factors could influence the outcomes of each procedure. For instance, MRI assessment and grading, as well as the contouring of the suspected lesions, could be influenced by the individual radiologist's skill and expertise, even when employing image-evaluating protocols such as the PIRADS scoring system with the aim of standardizing techniques and interpretation across centers. The employed technology also plays a major role in performing the biopsy and could influence the outcomes of each procedure, thus making it difficult to evaluate only the individual surgeon's skills. Another possible limitation is that there was a higher number of PIRADS 4–5 in group 1 when compared to group 2, which may be considered a source of bias in the analysis. However,

the difference is very close to being not statistically significant ($p = 0.040$), and the study was not designed to test these differences. Notwithstanding these limitations, our study is the first RCT comparing trained residents and consultants when performing fusion biopsies.

## 5. Conclusions

We can conclude that, after undergoing adequate training, and thus reaching the plateau of the learning curve, residents have similar outcomes in performing fusion prostate biopsies when compared to expert senior urologists. The exact role of surgical expertise in prostate biopsies is still to be determined, especially in those challenging cases with very small lesions. Future studies should carefully assess these aspects.

**Author Contributions:** Conceptualization, C.D.N., A.C. (Antonio Cicione), A.T., R.L. and A.L.P.; methodology, R.L., C.D.N., A.L.P., A.C. (Antonio Carbone), G.F. and A.F. (Antonio Franco); validation, R.L., C.D.N., A.C. (Antonio Cicione), A.T. and A.F. (Antonio Franco); formal analysis, R.L., A.F. (Antonio Franco) and A.C. (Antonio Cicione); investigation, B.T., G.T., A.C. (Antonio Carbone), A.F. (Andrea Fuschi) and G.T.; resources, A.T. and A.N.; data curation, A.L.P., A.N., A.T., A.F. (Antonio Franco), A.F. (Andrea Fuschi), G.T., R.L. and B.T.; writing—original draft preparation, B.T., R.L., C.D.N. and A.L.P.; writing—review and editing, R.L.; A.F. (Antonio Franco); visualization, B.T. and A.N.; supervision, C.D.N., A.C. (Antonio Carbone), A.F. (Antonio Franco) and G.T.; project administration, A.L.P., C.D.N. and A.T. All authors have read and agreed to the published version of the manuscript.

**Funding:** This research received no external funding.

**Institutional Review Board Statement:** The study was approved by the local ethics committee: IRU study—Prot. n. 258 SA_2021.

**Informed Consent Statement:** Informed consent was obtained from all subjects involved in the study.

**Data Availability Statement:** The data presented in this study are available on request from the corresponding author.

**Conflicts of Interest:** The authors declare no conflicts of interest.

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
