# Peer review of "Residents and Consultants Have Equal Outcomes When Performing Transrectal Fusion Biopsies: A Randomized Clinical Trial"

_curroncol, doi:10.3390/curroncol31020055_

Round 1

Reviewer 1 Report (Previous Reviewer 1)

Comments and Suggestions for Authors

The manuscript has been improved

Reviewer 2 Report (Previous Reviewer 2)

Comments and Suggestions for Authors

Accepted in the revised version.

This manuscript is a resubmission of an earlier submission. The following is a list of the peer review reports and author responses from that submission.

Round 1

Reviewer 1 Report

Comments and Suggestions for Authors

The manuscript is interesting but some points should be improved

1. In the Abstrac and in the Results PIRADS score >3 should be changed (I think) in PIRADS score  => 3; at the same time csPCa is a PCa ISUP Grade Group => 2

2. The location and size of suspicious mpMRI lesions should be reported (peripheric vs anterior zone)

2. The Greatest percentage of cancer or the lenght in millimeters of PCa cores should be reported

3. The patients with a PIRADS score 4 and 5 are significantly higher in Group 1 (64.3%) in comparison with Group 2 (40%); these data should be clearly reported in the Discussion

4. The role of transperineal prostate biopsy recommended by EAU guidelines because decrease the risk of sepsis should be reported (Pepe P, Pennisi M. Morbidity following transperineal prostate biopsy: Our experience in 8.500 men. Arch Ital Urol Androl. 2022 Jun 29;94(2):155-159. doi: 10.4081/aiua.2022.2.155. PMID: 35775338)

5. The higher detection rate of tranperineal targeted biopsy in comparison with transrectal approach in the diagnosis of anterior prostate cancer should be reported in the Discussion

Author Response

Reviewer #1:

Comments:

  1. In the Abstract and in the Results PIRADS score >3 should be changed (I think) in PIRADS score => 3; at the same time csPCa is a PCa ISUP Grade Group => 2.

We thank the reviewer for his/her comment and for the possibility to improve our manuscript.

The suggested modifications were made. PIRADS score >3 was corrected to PIRADS score ≥3 as well as csPCa was corrected from ISUP GG >2 to ISUP GG≥2, as conceptually originally intended by the authors.

  1. The location and size of suspicious mpMRI lesions should be reported (peripheric vs anterior zone).

We thank the reviewer for his/her comment and for the possibility to improve our manuscript.

We have added this data in the tables.

  1. The Greatest percentage of cancer or the lenght in millimeters of PCa cores should be reported.

We thank the reviewer for his/her comment and for the possibility to improve our manuscript.

We have added Max cancer length in the tables.  

  1. The patients with a PIRADS score 4 and 5 are significantly higher in Group 1 (64.3%) in comparison with Group 2 (40%); these data should be clearly reported in the Discussion.

We thank the reviewer for his/her comment and for the possibility to improve our manuscript. We agree that there is a significant difference between groups in these terms. We agree with the reviewer that this may be considered a source of bias in the analysis and therefore we have updated the limitations section. It is important to underline that the difference is very close to be not statistically significant (p= 0,040) and our study was not designed to test this difference.  

See Limitations section:

Another possible limitation is that a higher number of PIRADS4-5 in group 1 when compared to group 2 which may be considered a source of bias in the analysis. However, the difference is very close to be not statistically significant (p=0,040) and the study was not designed to test these differences.

  1. The role of transperineal prostate biopsy recommended by EAU guidelines because decrease the risk of sepsis should be reported (Pepe P, Pennisi M. Morbidity following transperineal prostate biopsy: Our experience in 8.500 men. Arch Ital Urol Androl. 2022 Jun 29;94(2):155-159. doi: 10.4081/aiua.2022.2.155. PMID: 35775338)

We thank the reviewer for his/her comment and for the possibility to improve our manuscript.

The role of transperineal prostate biopsy, highlighting the decreased rates of sepsis and subsequent hospital re-admission, according to the EAU guidelines and to the suggested literature, was introduced in the manuscript.

See Discussion:

According to the EAU guidelines transperineal should be preferred over transrectal biop-sies to improve cancer detection rate and to reduce risk of infections and sepsis (sepsis rates of 0.1% and 0.9% for transperineal and transrectal biopsies respectively) [7], thus lowering the risk of subsequent hospital re-admission. According to Pepe et al, on a series of 8.500 men undergoing a transperineal prostate biopsy, 1.5% of them required hospital re-admission due to complications but none developed sepsis [35].

  1. The higher detection rate of tranperineal targeted biopsy in comparison with transrectal approach in the diagnosis of anterior prostate cancer should be reported in the Discussion.

We thank the reviewer for his/her comment and for the possibility to improve our manuscript.

The higher detection rate of TP targeted biopsy in the diagnosis of anterior PCa was added in the manuscript according to data in literature, as suggested.

See Discussion:

A recent systematic review from Uleri et al found no statistically significant difference in the detection of csPCa between TR and TP approaches but, when stratified by lesion location, the TP approach was associated with higher csPCa detection of anterior and apical lesions [39].

Reviewer 2 Report

Comments and Suggestions for Authors

Dear authors,

This is an interesting article, but there are some issues to be resolved before a possible publication.

- a 10 days antibiotic scheme for a prostate biopsy is not recommended by the EAU Guidelines. So, the results of a non infectious procedure is probably biased

- please provide the kind of the administered antibiotics

- please provide data about the study statistical power

- you compare cancer detection rates between different patients. This comparison could provide safe results only if you were able to compare these rates on the same patient

- those trainees with less experience, performing only random biopsies should be excluded. Please, provide new data and results with revised number of urologists

- please be less definite with your conclusions. Your recruitment is small and there are quite few limitations.

Author Response

Reviewer #2:

Comments:

  1. A 10 days antibiotic scheme for a prostate biopsy is not recommended by the EAU Guidelines. So, the results of a non infectious procedure is probably biased.

We thank the reviewer for his/her comment and for the possibility to improve our manuscript. Our antibiotic prophylaxis is of five days we apologize for the typo error. Moreover, as correctly observed, the EAU guidelines do recommend antimicrobial prophylaxis to significantly reduce infections after transrectal prostate biopsy, suggesting the use – among others – of cephalosporin for 3 days starting 24 hours before the procedure. We considered it worthwhile to take into consideration other factors when defying the most appropriate antibiotic prophylaxis to be administered to our patients, such as the local antimicrobial resistance and our hospital antibiotic stewardship protocol. So far, to compliance our hospital guidelines on UTI prevention and treatment we have developed a protocol for prostate biopsy which has been evaluated in several trials published by our group. Just recently Quinolones have been replaced by sulfamethoxazole and trimethoprim after the EMA warning on quinolones.

See New references:

Lombardo R, Tema G, Nacchia A, Mancini E, Franco S, Zammitti F, Franco A, Cash H, Gravina C, Guidotti A, Gallo G, Ghezzo N, Cicione A, Tubaro A, Autorino R, De Nunzio C. Role of Perilesional Sampling of Patients Undergoing Fusion

Prostate Biopsies. Life (Basel). 2023 Aug 10;13(8):1719.

DE Nunzio C, Nacchia A, Lombardo R, Franco A, Cicione A, Trucchi A, Labella M, Bartoletti R, Simonato A, Ficarra V, Tubaro A. Is EMA warning on quinolones and fluoroquinolones really assessed? An EudraVigilance database analysis.

Minerva Urol Nephrol. 2023 Jun;75(3):374-380.

De Nunzio C, Presicce F, Lombardo R, Cancrini F, Petta S, Trucchi A, Gacci M, Cindolo L, Tubaro A. Physical activity as a risk factor for prostate cancer diagnosis: a prospective biopsy cohort analysis. BJU Int. 2016

Jun;117(6B):E29-35. doi:

De Nunzio C, Tema G, Trucchi A, Cicione A, Sica A, Lombardo R, Tubaro A. Smoking reduces PSA accuracy for detection of prostate cancer: results from an Italian cross-sectional study. Minerva Urol Nefrol. 2019 Dec;71(6):583-589.

De Nunzio C, Lombardo R, Nacchia A, Tema G, Tubaro A. Repeat prostate- specific antigen (PSA) test before prostate biopsy: a 20% decrease in PSA values is associated with a reduced risk of cancer and particularly of high-grade

cancer. BJU Int. 2018 Jul;122(1):83-88. doi: 10.1111/bju.14197. Epub 2018 Apr 10. PMID: 29533522.

See Materials and Methods – Study population

Prior to the procedural date, due to hospital protocols and according to local antimicrobial resistance, patients were administered prophylactic antibiotic therapy with oral sulfamethoxazole and trimethoprim (800mg + 160 mg; 2 tabs day)  for three days before the biopsy and two days after and Amikacine 500 mg i.v periprocedurally. [27].

  1. Please provide the kind of the administered antibiotics

We thank the reviewer for his/her comment and for the possibility to improve our manuscript.

We integrated the manuscripit providing the administered antibiotic.

See Materials and Methods – Study population

Prior to the procedural date, due to hospital protocols and according to local antimicrobial resistance, patients were administered prophylactic antibiotic therapy with oral sulfamethoxazole and trimethoprim (800mg + 160 mg; 2 tabs day)  for three days before the biopsy and two days after and Amikacine 500 mg i.v periprocedurally. [27].

  1. Please provide data about the study statistical power

We thank the reviewer for his/her comment and for the possibility to improve our manuscript.

Data on statistical power is present in the statistical section of the manuscript:

‘Prior to the study start, a power calculation was performed, based on the assumption of a better performance of 10% for consultants, when compared to residents, in terms of csPCa detection. To identify such a variation, it was estimated that 140 evaluable patients were needed, with 80% power using a two-tailed test at 5% significance level.’

  1. You compare cancer detection rates between different patients. This comparison could provide safe results only if you were able to compare these rates on the same patient

We thank the reviewer for his/her comment and for the possibility to improve our manuscript

We thank the reviewer for his/her comment. We agree that the ideal design of the study would be to perform biopsies on the same patients for both groups however it is clear that several ethical issues arise with such a design. We performed an adequately powered study to prove that after adequate training residents can safely perform fusion biopsies with similar detection rates and safety compared to consultants. As with any other surgical procedure, expertise should not be diminished, and a study is ongoing to analyze the ability of different urologist with different expertise in correctly sampling small lesions. Small lesions in very large prostates represent a challenging procedure particularly when located on the base of the prostate peripherically. In order to improve the manuscript we have improved the discussion section.

See Discussion section:

‘The ideal design to compare the detection rate between both groups would have been to biopsy the same patients in both groups. However, such a design, presents important technical and ethical issues. The present study confirms trained residents reach comparable results when compared to consultants, however it is important to underline the importance of surgical expertise. A study is ongoing in our center to explore the importance of surgical expertise in patients with small PIRADS 4 lesions (<10mm). Small lesions in the base of the prostate peripherically represents a challenge for most clinicians performing fusion biopsies.

  1. Those trainees with less experience, performing only random biopsies should be excluded. Please, provide new data and results with revised number of urologists

We thank the reviewer for his/her comment and for the possibility to improve our manuscript.

The random biopsies were performed by the trainees only during the biopsy training, thus not considered into this study.

The biopsies included in the study are the ones performed by the residents only after undergoing the required training, that consisted of 50 procedures.

We reformulated the sentence to make it clearer to the reader.

See Materials and Methods – Study population

Residents were considered “trained” after performing at least 50 procedures prior to the beginning of the study. The training consisted of the first 20 procedures in which the resident performed only the local anesthesia and the random biopsies, while in the following 30, the resident performed the whole procedure. Training was carried out by a senior urologist with more than 10 years of experience. All the biopsies performed by trained residents that we considered in the present study were conducted only after the resident underwent such training, thus the training bioptic procedures are not included in the study.

  1. Please be less definite with your conclusions. Your recruitment is small and there are quite few limitations.

We thank the reviewer for his/her comment and for the possibility to improve our manuscript. Conclusion has been rewritten.

See New Conclusions:

We can conclude that, after undergoing adequate training thus reaching the learning curve, residents have similar outcomes in performing fusion prostate biopsy when compared to expert senior urologists. The exact role of surgical expertise in prostate biopsies is still to be determined, especially in those challenging cases with very small lesions.  Future studies should carefully assess these aspects.